# Sociodemographic Determinants of Poles’ Attitudes towards the Forest during the COVID-19 Pandemic

**DOI:** 10.3390/ijerph19031537

**Published:** 2022-01-29

**Authors:** Anna Koprowicz, Robert Korzeniewicz, Wojciech Pusz, Marlena Baranowska

**Affiliations:** 1Institute of Pedagogy, Pomeranian University in Słupsk, 76-200 Słupsk, Poland; anna.koprowicz@apsl.edu.pl; 2Department of Silviculture, Poznan University of Life Sciences, 60-637 Poznań, Poland; robert.korzeniewicz@up.poznan.pl; 3Department of Plant Protection, Wrocław University of Environmental and Life Sciences, 50-375 Wrocław, Poland; wojciech.pusz@upwr.edu.pl

**Keywords:** coronavirus, forest therapy, LAS scale, COVID-19, society, forest function, urban and suburban forests

## Abstract

Attitudes towards forest ecosystems have been changing together with human needs, which is amplified with society’s increasing need to spend recreation time in the forest. The phenomenon has been particularly visible during the COVID-19 pandemic. The aim of this study was to determine the attitude of Poles to forests during the COVID-19 pandemic. The research was based on (1) a sociodemographic background questionnaire that consisted of questions about the independent variables and (2) the LAS scale—an independently prepared tool for measuring attitudes towards the forest. In the survey, 1025 people participated (673 women). The age of the subjects was between 19 and 68. The attitude towards the forest was analysed in three dimensions: Benefits, Involvement, and Fears. The Mann–Whitney U test and Kruskal–Wallis one-way analysis of variance by ranks were used for statistical analysis. Women and people with primary education expressed the most fears connected with going to the forest. Men and people living in the countryside and in small towns, as well as respondents who were professionally active and performing work connected with forests were the most involved in exploring the forest and working for its benefit. Concerning the forest, concerned women, people from the highest age group, respondents with university education, and white-collar workers notice the most benefits from recreational activities in the forest.

## 1. Introduction

### 1.1. Forest and Health

The World Health Organisation (WHO) has acknowledged forest ecosystems as key for the survival of the humankind and the life on Earth [1]. The perception of forest ecosystems has been changing together with human needs [2,3]. It has been estimated that forests fulfil over 100 functions that are perceived as benefits which humans can accrue. The most popular benefits are economic, social, and protective [4,5]. The social benefit of the forest, which is constantly gaining importance, is to create optimal conditions for human health and recreation [5,6]. The modern person in an anthropogenic environment, who is constantly pursuing free time to regenerate both the body and mind, is spending an increasing amount of time in the forest [7,8].

The idea of spending time in a natural environment for its regenerative, restoring, and healing properties has been known since the 16th c. In Europe, people suffering from breathing difficulties, tuberculosis, and some mental illnesses were directed to health resorts surrounded by forests [9]. Nowadays, so-called forest therapy and forest bathing are becoming more popular [10,11,12], and involve walks in the forest in each period (a weekend, a week, or longer) depending on the ailments and the person’s health [13]. This concept was introduced in Japan in 1982 by the Forestry Agency, and since then it has been gaining popularity, particularly among corporate employees where a significant number of worker deaths had been observed [14]. Nowadays, these methods are used in other countries as well [10,15]. The medical community has welcomed forest therapy as a preventive treatment [13] and it has been used to assist in the recuperation and rehabilitation process. The therapy aims at stimulating the body to self-heal through contact with nature in the forest. It has also been used in treating disorders that occur due to stress, depression, and ageing, and to improve overall health and wellbeing (the rustling sounds of trees, bird song, and the greenery have a soothing effect). Walking in the forest has also been suggested to humans as a form of minimal physical activity, getting fresh air (oxygenate the body), and improving blood circulation [10,11,14,16]. Moreover, spending time in the forest distances people from places where they usually spend time, e.g., the workplace, and distracts them from everyday activities, reduces stress levels, and improves mood and concentration [17]. Forest bathing has a positive influence on emotions and regeneration, and increases the level of vitality, which has been emphasised by Karjalainen et al. [18] and Bielinis et al. [19].

Healing properties of the forest stem from its microclimate, which is shaped by essential oils (phytoncides). They are recognised for their antibacterial, fungicidal, and antiviral properties [20]. Hence, forest therapy should be popularised in society, especially in many places on the planet where forests are easily accessible. Walking in the forest is a cheap leisure activity that is of vital importance for elderly people and the ill whose economic situation is precarious [16].

Despite numerous benefits that the forest offers, there are also people who do not like the forest and perceive it as an alien and dangerous place, evoking anxiety and uneasiness [21]. In some situations, the forest ecosystems can be threatening to human health. People who are in frequent contact with the forest can be exposed to infectious diseases connected with it. The vectors of these diseases can be the arthropods and mammals living in the forest. Examples of such diseases are *Puumala orthohantavirus* (PUUV, transmitted by the bank vole), Lyme disease, and tick-borne encephalitis (transmitted by ticks) [22]. Another why the forest evokes fear are dangerous animals, as well as poisonous plants and mushrooms that can be found there. Also, some animals and plants which are part of the forest ecosystems cause allergies and skin reactions [18].

### 1.2. The COVID-19 Pandemic and the Importance of the Forest

Since the beginning of the COVID-19 pandemic (in April 2020) it was evident that the situation was going to impact forests and forestry [23]. In Poland, the first case was diagnosed on 4 March 2020, while the first regulations to limit the spread of the coronavirus were introduced on 13 March 2020. Regulations [24] limited the possibilities of movement, the functioning of certain institutions or workplaces, and established a ban on assembly. So far, these restrictions have appeared in Poland depending on the number of infected people. The State Forests National Forest Holding (PGL LP) also introduced a periodic ban on access to the forest on 3–11 April 2020, under the slogan “Las Poczeka” (the forest will wait). After that period, parks and forests were often the only “space of freedom” for Poles with unlimited access [25]. Forests in Poland cover about 30% of the country’s area, of which almost 81% are public forests. These are national goods, owned by all citizens of the Republic of Poland, to which every Pole has free access. [26]. Currently, the distance of the place of residence from the forest and the possibility of access are not a factor that significantly affects the limitation of recreation in forests [25].

Derks, Giessen, and Winkel [27] compared the number of visitors to forests in Germany before and during the introduction of restrictions and mandates against COVID-19. The scientists concluded that the numbers of visitors to the forests doubled. People were also more motivated to go to the forest for social reasons, such as meeting friends and family, as well as for preserving physical and mental health [28]. A similar increased interest in forest mid-COVID-19 pandemic was observed by Grima at al. [29]. The increasing trend in forest visits during the COVID-19 lockdown in 2021, compared to the same months before the pandemic in 2016 and 2017, was written about by Bamwesigye et al. [30]. Many other scientists noted an increased number of people who strolled and rode bikes along rivers or in parks and forests during the lockdown [31,32]. During the pandemic, people pay special attention to the risk of infection, and the forest, unlike public space, is not a crowded place. Especially at the times when recreation, sport, and cultural facilities, as well as shopping centres, were closed, the forest was one of few free spaces where Poles could spend their free time [25]. The pandemic made Poles change summer holiday plans [33,34], both due to travelling restrictions and deteriorated financial situations [34], particularly for people whose business activity suffered because of the restrictions. The majority of people had to limit their expenses on recreation [9]. The pandemic functioned as an inhibitor for the tourist industry [35,36] but it also evoked the substitution effect on the tourist market [33]. One of the substitutes for domestic and international tourism could be trips to the forest. During the pandemic, the forest can be a particularly popular place for walks, considering the necessity to avoid crowds as well as the opportunity to find peace and relaxation from everyday life [25]. In such a situation, it is important to notice that spending time in the forest on a regular basis boosts the immune system [9,20] by mitigating the effects of social isolation and loneliness, which have a negative impact on both physical as well as mental health [37].

## 2. Materials and Methods

The increase in the number of people visiting forests is a challenge for those who manage forests as well as for urban forest policy [27]. The boom of visitors generates the need for integrated forest management [38] that can respond directly to the social need and will be suited to people’s needs and expectations. The strategy should be based on the understanding that management measures are essential to deliver various ecosystem services needed by society [27]. In this context, it seems especially important to canvass popular opinion about the forest and forest recreation. And this topic was increasingly taken up by researchers, such as Bamwesigye et al. [30], Mateer et al. [39].

### 2.1. The Aim, Problem, and Hypothesis of the Research

Considering the above ideas, the aim of this study is to learn about the attitude of the Polish people towards the forest regarding chosen sociodemographic variables. The aim of the study was to determine the attitude of Poles towards the forest amid the COVID-19 pandemic. The pilot studies conducted by Baranowska, Koprowicz, and Korzeniewicz [25] reveal that during the restrictions, to prevent the spread of COVID-19, the forest gained significance as one of few spaces of freedom for Poles. The study participants indicated that the lower transmission risk of SARS-CoV-2 virus is one of factors which encouraged walks. Hence, it is necessary to inquire about a general attitude of the Polish people towards the forest amid the pandemic and what are the characteristics of this attitude.

The following research question has been formulated: What are the sociodemographic factors of Poles’ attitude towards the forest? The dependent variable is the attitude towards the forest expressed by results of LAS scale which includes three areas: the Benefits of forest recreation, the fears connected with the forest, and involvement in exploring the forest. The independent variables controlled in the study are sex, age, professional activity, work, education, place where the respondents live, and their living conditions.

We formulated several hypotheses: (1) women express more fears connected with going to the forest then men, (2) elderly people see more benefits connected with the forest than younger respondents, (3) people with secondary and higher education express fewer fears connected with being in the forest and see more benefits, (4) people living in the country see more benefits of forest recreation and are more involved than people living in the cities, (5) people living in a block of flats see more benefits of being in the forest than people who have their own gardens, (6) people who are professionally active see more benefits connected with the forest, but they are also less involved than people who do not work, and (7) people who work in professions connected with forest management are more involved and express fewer fears connected with the forest than people who work in other professions.

### 2.2. Research Tools

The study was conducted using the survey diagnostic method and two research tools were used:Sociodemographic background questionnaire that consisted of questions about the independent variables.LAS scale—independently prepared tool for measuring the attitude towards the forest. The tool is characterized by good reliability expressed by Cronbach’s alpha coefficient at the level of 0.90. Its theoretical relevance was determined by subjecting the items to competent judges and by factor analysis. The scale contains 20 statements, to which the respondents refer on a scale from 1—“strongly disagree” to 5—“strongly agree”. It consists of three factors. The first one, “Benefits”, consists of eight statements (such as: “I rest and relax in the forest very well”, “The forest helps me to improve my health”) concerning the experience of pleasure and health benefits connected with spending time in the forest. The second factor, “Involvement”, included eight statements that determine the extent to which respondents are involved in learning about nature, understanding forest management issues, or working for the benefit of the forest. There were such statements as: “I take part in tree planting campaigns”, “I am interested in nature—I am keen on learning about various tree species or reading about animals”. The last factor, “Fears”, containing four statements (e.g., “I am afraid to go to the forest because of ticks” or “I worry that I will get lost in the forest”), examines the concerns expressed by the respondents about being in the forest.

### 2.3. The Research Procedure

The results presented in the article are a part of a greater interdisciplinary study titled “The approach towards the forest during the pandemic. Psychological and sociodemographic characteristics”. The analysed research tools, together with others that are a part of the entire project, were shared on the Internet between February and May 2021. The invitation to participate in the study was sent through social media. Students were asked by their academic teachers who work at higher education institutions of various profiles. Out of the 1071 questionnaires that were sent, 1025 were qualified.

Statistical calculations were conducted using Statistica 10 software (StatSoft Polska Sp. z o.o., Kraków, Poland). The distribution of the results achieved in the LAS scale deviates from normal, hence, to determine the differences between the particular groups, the Mann–Whitney U test and Kruskal–Wallis one-way analysis of variance by ranks were adopted.

### 2.4. Surveyed People

In the study (survey), 1025 people participated, out of which 673 were women. The ages of the subjects were between 19 and 68. To conduct comparisons, the respondents were divided into three age groups. The aim was to create groups that were similar in terms of numbers, hence percentiles were adopted, with the lower threshold established at 33.34% and the upper threshold at 66.66%. Thus, people up to the age of 22 were assigned as the youngest group, and people over the age of 26 were assigned to the oldest group. The detailed sociodemographic structure of the surveyed people is presented in Table 1.

## 3. Results

### 3.1. Sex and Attitude to the Forest

The results of the Mann–Whitney U test indicate statistically significant differences between males and females in the scope of all subscales in the LAS scale. In light of the collected research material, the first hypothesis can be considered as a supported one—females do indeed express more fears connected with going to the forest. Moreover, it turned out that the differences between males and females occur also in Benefits and Involvement (Table 2).

### 3.2. Age and Attitude towards the Forest

The results of Kruskal-Wallis test indicate a significant difference in the scope of noted benefits from forest recreation (H (2, *n* = 1025) = 31.45, *p* < 0.01); involvement in exploring the forest (H (2, *n* = 1025) = 34.74, *p* < 0.01), and fears connected with spending time in the forest (H (2, *n* = 1025) = 8.93, *p* = 0.011), which were signalled by various age groups. The differences are presented graphically in Figure 1.

Post-hoc analysis (Dunn’s test) helped to determine which of the three groups the differences in particular scales of the attitude to the forest have reached the necessary level of statistical significance (Table 3). On the “Benefits” and “Involvement”, scale all three groups differ significantly from one another. The oldest people in the study group (group III) recognize the most benefits coming from contact with nature, whereas the youngest group (group I) sees the fewest benefits. As far as “Involvement” is concerned, the respondents form the second age group achieved significantly higher results than the other ones. In the “Fears” scale, the statistically significant differences appeared between group II and group I, who expressed the most anxiety.

### 3.3. Education and Attitude towards the Forest

Statistical analysis has revealed that the respondents at various degrees of education differ as far as forest Benefits are concerned (H (3, *n* = 1025) = 23.91, *p* < 0.01). Such differences were not observed in the subscales focused on Fears H (3, *n* = 1025) = 2.48, *p* = 0.48) and Involvement (H (3, *n* = 1025) = 10.81, *p* = 0.012). The mean comparison showed that people with higher education most often acknowledge Benefits of spending time in the forest, whereas people with primary education—the least often (Figure 2).

The post-hoc analysis has revealed that a statistically significant difference in this scope occurs only between people with higher education and secondary education. Among other groups the difference was not statistically significant (Table 4).

### 3.4. Place of Residence and the Attitude towards the Forest

The result of the analysis has revealed that the type of settlement where the respondents live also differentiates their attitude to the forest as far as their Involvement is concerned (H (3, *n* = 1025) = 55.83, *p* < 0.01). The differences concerning Fears (H (3, *n* = 1025) = 5.14, *p* = 0.16) and Benefits (H (3, *n* = 1025) = 6.20, *p* = 0.10) appeared to be statistically insignificant. The collected means have been depicted in Figure 3.

The post-hoc analysis showed that residents of villages and small towns are more involved in learning about and exploring the forest and are more involved in working for their benefits than inhabitants of big towns and cities (Table 5).

The variable known as “type of dwelling” (living conditions) has recognized the following types of houses: a block of flats, a detached house with a garden, and multi-family houses with access to a garden. This variable also differentiated the respondents’ in reference to Involvement (H (2, *n* = 1025) = 18.85, *p* < 0.01), but it does not matter as far as Benefits (H (2, *n* = 1025) = 1.99, *p* = 0.37) or Fears (H (2, *n* = 1025) = 1.23, *p* = 0.54) are concerned. Statistically significant differences on the “Involvement” scale occurred between people who live in a block of flats and other groups (Table 6).

### 3.5. Professional Activity and Attitude towards the Forest

In the scope of declared professional activity, three groups were distinguished: professionally active, professionally inactive, and those whose activity ceased due to retirement. The variable appeared to be a factor differentiating the respondents as far as Benefits (H (3, *n* = 1025) = 14.48, *p* < 0.01), Involvement (H (2, *n* = 1025) = 26.5, *p* < 0.01), and Fears (H (2, *n* = 1025) = 9.48, *p* < 0.01) are concerned. At the same time, the post-hoc analysis shows that the differences reach a satisfactory level of statistical significance only when comparing people who are professionally active and inactive, but not when comparing with people who retired (Figure 4). People professionally active perceive more benefits connected with forest recreation than people professionally inactive; they are also characterized by highest Involvement in the studied group. Also, they express fewer Fears about spending time in the forest than professionally inactive people.

Type of occupation also appeared to be a significant variable to determine the respondents’ attitude towards the forest. It differentiates the respondents in reference to Benefits (H (7, *n* = 1025) = 15.483, *p* = 0.03) and Involvement (H (7, *n* = 1025) = 86.64, *p* < 0.01) as well as Fears H (7, *n* = 1025) = 32.23, *p* < 0.01). As expected, people whose occupation relates to the forest are characterized by an increased Involvement and fewer Fears than other respondents (Figure 5).

The assumptions were also corroborated by a post-hoc analysis. People professionally connected with the forest in reference to Fears do not differ statistically significantly from uniformed services and blue-collar workers (physical workers) and services. As far as Involvement is concerned, they differ from white-collar workers, freelancers and services, and health workers, but they do not differ from uniformed services and blue-collar workers (physical workers). In this area, blue-collar workers differ from other groups, apart from forest workers and uniformed services. In the Benefits subscale, the statistically significant differences occurred only between professionally inactive people and white-collar workers. People working in forestry do not differ from people of other types of occupations in the assessment of benefits connected with forest recreation

## 4. Discussion

Despite the fact that it is women who feel more benefits connected with spending time in the forest and forest recreation, men are more involved both in the work in forest areas as well as in exploring nature. This has also been confirmed by studies done by Gołos [7], who claimed that women who went to the forest more often also paid more heed to the benefits connected with relaxation, learning, and educational processes, as well as mood enhancement. Men, however, revealed a greater need for physical activity and a need to satisfy their curiosity (seeking attractions) [40]. Women express greater fears of encountering wild animals and ticks [41]. Hence, a trip to the forest with family and friends can be important for women because they do not always want to go to the forest alone, for safety reasons [42]. Studies from previous years (e.g., [43,44,45]) reveal that women and people with higher education are more often characterized by ecological sensitivity than other groups. Nevertheless, Trempała and Sadowski [46], who studied social attitudes towards deforestation of tropical forests, proved that men were characterized by a higher level of increasing biocentric attitudes than women. This view has also been confirmed by our studies which focused on greater involvement and willingness of men to work to forests’ benefit.

Hypothesis No. 2, which stated that older people recognize more Benefits connected with the forest, has been confirmed. Indeed, the respondents from the oldest age group are significantly statistically different in comparison to other studied groups. Moreover, the conducted research reveals that age is a differentiating factor as far as Involvement is concerned. On this scale, the highest results were achieved by the people ascribed to the middle age group and this group also differs significantly from the youngest group in reference to declared Fears. One of the factors that encourages people to visit nearby green areas and forests is the willingness to socialize. Short social contacts in the forests were identified as, for example, walks with a dog and other forms of leisure activity in the forest with other people [47]. Participating in events that take place in forests can be a source of social contact for people. Considering the benefits of spending time in the forest for people, it seems rational to promote this type of leisure activity. A similar idea was presented by Zawadka and Zawadka [16], particularly in reference to recreation and forest therapy. It is important to pay special attention to the oldest people (over 65 years old) and research should be conducted in this group because it is claimed that by 2060, the number of people aged 65 and more will have increased up to 30% in Europe, and the share of people above the age of 80 will also increase [48]. Hence, it is reasonable to make changes and preparations in the forest for the rest and recreation of elderly people and the disabled [49].

The third hypothesis assumed that people with higher and secondary education have fewer Fears connected with the forest and acknowledge more Benefits. This hypothesis has been only partially confirmed, particularly in reference to Benefits. Interestingly, respondents with secondary education differ as far as Benefits are concerned only from the respondents with higher education. The differences between people from other groups dividing people by education were not found. Although it was visible that the mean of fears is the highest in the group with primary education, the differences appeared statistically insignificant. Tyrväinen et al. [50] pointed to the differences concerning the way nature is perceived through the prism of age, health, status and psychological features, and physical activity and education. In addition, Wierzbicka, Krokowska-Paluszak, and Schmidt [51] reported that most tourists who visited Przemęt Landscape Park were people with secondary or higher education. They stated that better-educated people are more willing to spend leisure time outdoors; it may be due to the fact that they have a greater awareness of the need to rest than those with a lower education [51], which also confirms our findings. According to Grzelak-Kostulska and Hołowiecka [52], with higher education, the awareness of the need to rest, especially as an active recreation, increases. The level of education/projects to the need of spending leisure time actively in a natural environment, which is beneficial to health, is realized by the people with higher education [52]. A partial corroboration of the third hypothesis is quite surprising because the research into the level of self-assessment of wellbeing during the pandemic revealed that education does not depend on it. The lower a respondent’s education, the more pessimistic they were in their questionnaire answers. Simultaneously, people with higher education indicated the need to give up recreation more often [53].

In the study, the focus on the place of residence was twofold: both in reference to the type of settlement as well as the type of house (dwelling) in which the respondents live. The living conditions became particularly significant during lockdown to prevent the spread of COVID-19 and also for people who were quarantined or who underwent home isolation. It seems that people whose movement was restricted were forced to stay indoors in a block of flats; naturally these groups could particularly feel the loss of relaxation in the fresh air. People who have a garden were able to catch their breath and relax during lockdown.

The hypothesis No. 4 has been confirmed only partially. Indeed, people who live in the country depict greater involvement than those who live in big towns and cities, although they do not differ in this scope from people living in small towns. The assumption concerning Benefits has not been confirmed; the respondents from cities do not differ from people living in the country in this scope. Forest penetration and the choice of recreation place are dependent on the location of the forest in reference to the place of residence and the preferred type of recreation chosen by the respondents [54]. Dąbrowski and Zbucki [41] did not notice differences in limitations concerning forest recreation due to the respondents’ place of residence. Gołos [7] indicates that people living in cities complain about diminishing recreation areas, including forest and areas with trees (parks). People living in the rural areas, to a great extent, do not experience the deficit of forest areas because it is an inalienable element of their surrounding and lifestyle. Moreover, forests are also a source of additional income for those who live in the country [7].

The predictions expressed in hypothesis No. 5 appeared to be erroneous. It was assumed that during the pandemic, people living in a block of flats would appreciate the opportunity to walk in the forest the most. The assumption was not confirmed. However, the conducted research has confirmed that people living in block of flats are less involved in exploring the natural environment than people who have the possibility of tending to their garden on a regular basis. It is quite surprising because blocks of flats are built most often in cities. The previous research indicates that spending time in the forest can lower the impact of these factors on the human body [55,56]. Hence, the benefits of spending time surrounded by nature should be appreciated by people living in blocks of flats, particularly during COVID-19 lockdowns.

The hypotheses concerning professional activity assumed, among others, that people professionally active acknowledge more benefits, but are also less involved than people professionally inactive. The hypothesis was confirmed in reference to Benefits, however the assumption about lesser Involvement was verified negatively. It would seem that people professionally active have less time that could be spend in order to get involved in forest exploration. However, it occurred that it is the opposite—it is those professionally active who are involved the most. A possible explanation could be that people who work seek isolation from the workplace, everyday chores, and activities. Our results confirm the claims of Pietrzak-Zawadka and Zawadka [16], who indicate that health tourism, which is particularly popular among professionally active people, is developing dynamically nowadays.

Hypothesis No. 7, which assumed the occurrence of differences between people working in forestry and forest management and respondents in the scope of Involvement and Fears, has been positively verified. Interestingly, as far as Involvement is concerned, people professionally linked with forests do not differ statistically significantly from people who work in uniformed services and blue-collar workers; and as far as Fears are concerned, apart from the two groups they also differ from people working in services. It seems that the explanation of the results can be partially found in the procedure of the conducted research. The respondents themselves had to choose their own type of occupation (explanations and lists which occupations and jobs are included in each type were provided). Unfortunately, the study conducted over the Internet did not provide the opportunity to solve any possible doubts and inquiries about the adopted types of occupations. There is a possibility that some people who work in forestry and forest management were classified as blue-collar workers, i.e., physical workers.

## 5. Conclusions

The relationships between health and wellbeing, biodiversity, healthy ecosystems, and climate change have been attracting attention of researchers and politicians internationally over the last few years [1,57,58]. The current situation around the world added one more issue to these ponderings, namely their relationships with the coronavirus pandemic [59].

A few years before the pandemic it had been observed that one of the most visited places were suburban forest complexes. It stems from the increasing awareness about the human need for recreation and, at the same time, an increased access to recreational places outside the city (among others transport, infrastructure) [52].

During the pandemic, the role of the forest, as a place for relaxation, recreation, and social gatherings, has increased. The living situation and personal features of character shape people’s attitude towards the natural environment [60]; hence we attempted to determine which sociodemographic variables are connected with Poles’ attitude towards the forest in the presented study. The relationship was analysed in three dimensions: Benefits, Involvement, and Fears. Most Fears connected with spending time in the forest were expressed by women and people with primary education, whereas men, people who live in the country and in small towns, people professionally active, and people working in forestry were the most involved in exploring the forest and working to its benefit. In the scope of observed Benefits of forest recreation, higher results were achieved by women, the elderly, and respondents with higher education; additionally, white-collar workers notice the benefits more often than professionally inactive people. Gathering opinions about the forest, including the sociodemographic characteristics, may be helpful in implementing forest practices that will meet contemporary social expectations [25]. Particularly, during the pandemic an increased number of visitors in the forest was a challenge for forest managers and for urban forest policy [27]. The increase in the number of visitors generates the need to integrate forest management [38], which will directly respond to social demand and will be suited to the needs and expectations of visitors. The strategy should be based on the understanding that management measures are essential to provide various ecosystem services, which society needs [27]. The results of further research will indicate the most socially acceptable ways of managing forests. They will contribute to proposing alternative methods in reference to, for example, currently used thinning methods, by replacing the clear-cutting method and selection-cutting method with the patch-selection method of forest management. The results of similar research will also support the decision concerning the organization of tourism in forests, i.e., canalizing tourist flow in the forest area, creating new parking spaces or creating an app, making it safer to use the forest.

Getting to know the needs and expectations of society in relation to the forest is important, because the demands of the society are often based on individual, subjective preferences [61]. Due to this fact, foresters should shape the attitudes of society in order to limit the habits of people that may pose a threat to nature or contribute to losses in forest management [7]. The results of social research make it possible to define the target scope and forms of forest recreation areas management, the evaluation of non-market values of the forest, as well as the benefits and risks resulting from multifunctional forestry.

## 6. Limitations

The adopted procedure of conducting the studies via the Internet made it possible to collect a considerable sample of people from various environments. Naturally, the approach has its limitations. One of the issues was representativeness: to take part in the study it, was necessary to have access to a computer and the Internet so the potential respondents could receive a link to the questionnaire; also, the topic of the study had to be interesting enough for people to devote their time and fill in the questionnaire. In such a case, it is difficult to infer a random choice of the sample, which would increase its representativeness and also the possibility to generalize the achieved results. Another vital value limitation of the adopted procedure is the possibility of filling in the questionnaire multiple times by the respondents or its absolute anonymity which can favour giving untrue data about sex or age, for example. Moreover, no possibility to control the external factors, such as distractions or the presence of third parties excluded the standard score of the research conditions [62]. The number of the collected sample lowers the unfavourable influence of the described factors on the achieved results; hence they can become an inspiration for further empirical studies.

The non-equivalence of the studied groups’ results in the necessity to treat some of the results cautiously. Despite adopting nonparametric methods of data analysis, some of the groups (such as the group of people working in uniformed services—3 people) are so small that it is difficult to generalize conclusions coming from comparisons with other groups.

## Figures and Tables

**Figure 1 ijerph-19-01537-f001:**
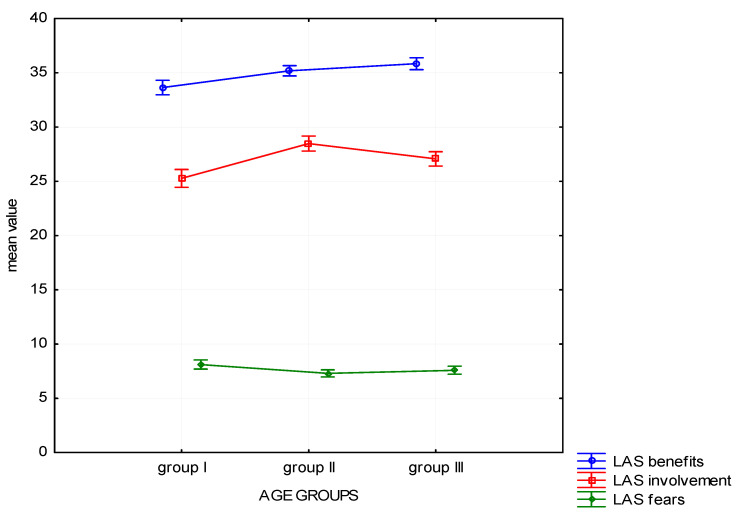
Attitude to the forest—mean chart for groups divided by age.

**Figure 2 ijerph-19-01537-f002:**
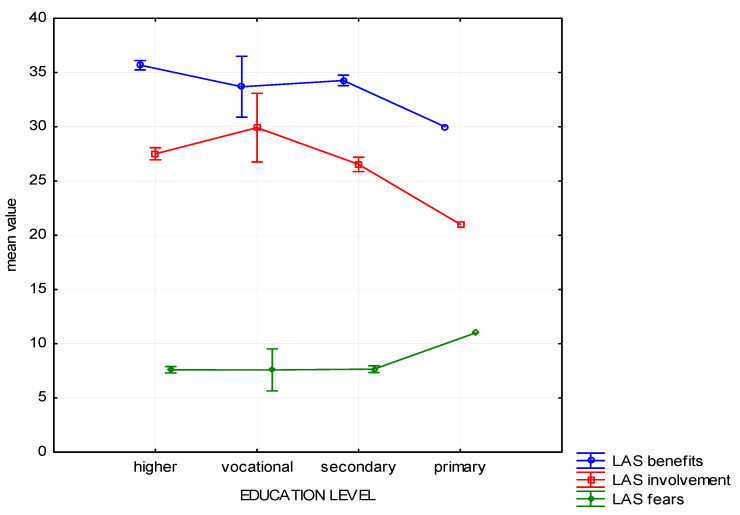
Attitude to the forest—mean chart for groups divided by education level.

**Figure 3 ijerph-19-01537-f003:**
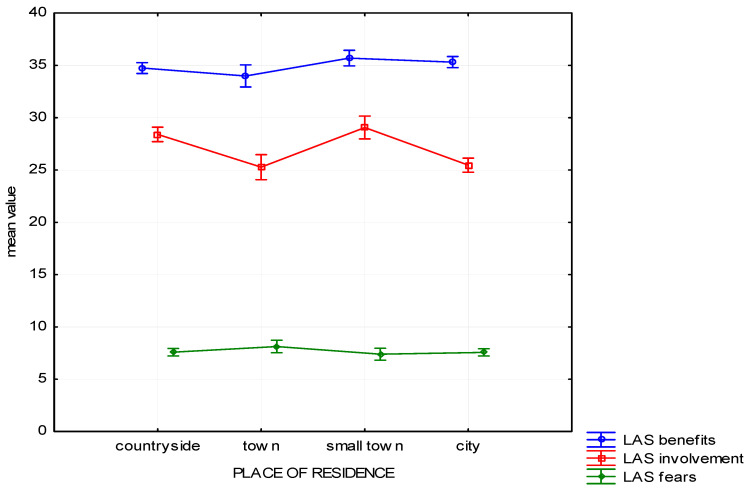
Attitude to the forest—mean chart for groups based on the place of residence.

**Figure 4 ijerph-19-01537-f004:**
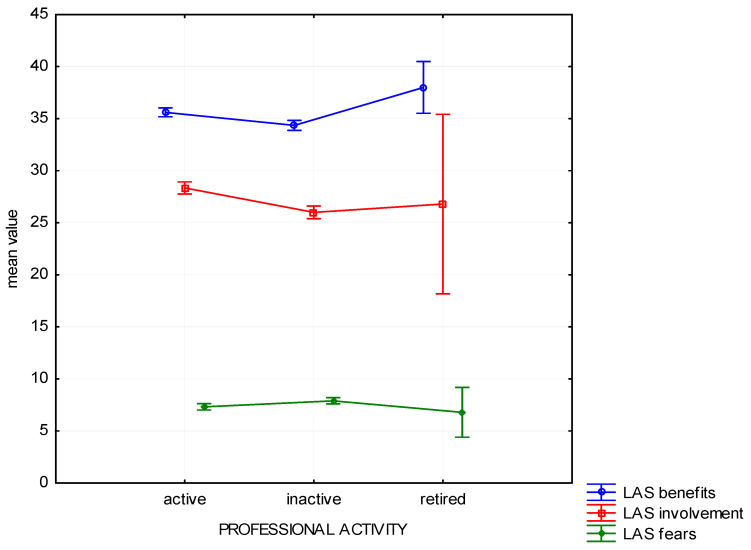
Attitude to the forest—mean chart for groups based on the place of residence.

**Figure 5 ijerph-19-01537-f005:**
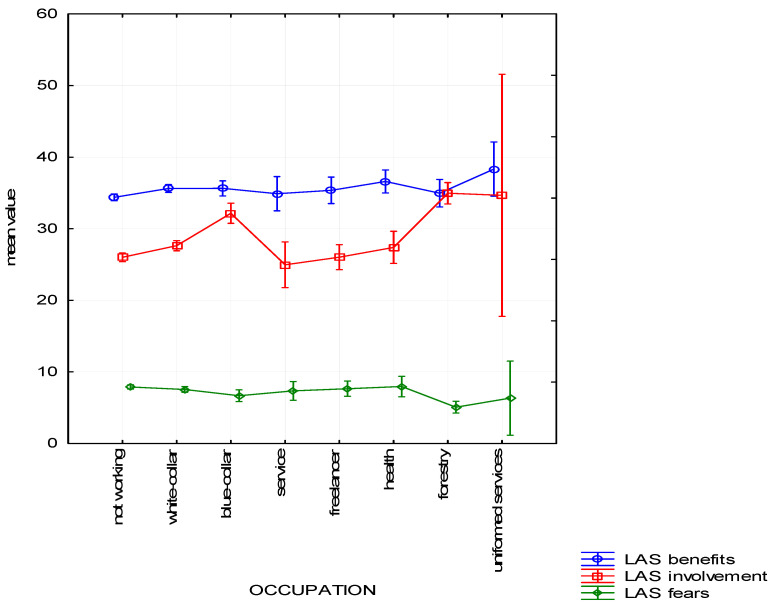
Attitude to the forest—mean chart for groups based on type of occupation.

**Table 1 ijerph-19-01537-t001:** Sociodemographic structure of the studied sample.

Variable	Feature	Share (%)
Sex	Male	65.66
	Female	34.34
Age	Group I (ages 19–21)	28.39
	Group II (ages 22–25)	39.02
	Group III (26 and over)	32.59
Education	Primary	0.09
	Vocational	2.54
	Secondary	46.83
	Higher	50.54
Place of residence/ Type of settlement	Countryside	40.00
	Small town (up to 20,000 inhabitants)	13.76
	Town (between 20,000 to 100,000 inhabitants)	13.37
	City (above 100,000 inhabitants)	32.88
Type of dwelling	Detached house with a garden	55.90
	Multi-family house with a garden	7.02
	Block of flats	37.07
Professional activity	Professionally active	47.22
	Professionally inactive	52.29
	Retired	0.49
Type of occupation	White-collar worker	29.76
	Blue-collar worker	5.66
	Service	2.73
	Freelancer	3.41
	Health worker	2.73
	Uniformed services	0.29
	Forestry	2.63
	Professionally inactive	52.78

**Table 2 ijerph-19-01537-t002:** Attitude to forest—comparing females and males.

LAS	Women			Men			U	*p*-Value
	*n*	Median	Quartile Range	*n*	Median	Quartile Range		
benefits	673	37	8	352	36	7.5	103,219	0.001
involvement		26	10		30	10	88,216	0.001
fears		8	6		5	3	70,892	0.001

**Table 3 ijerph-19-01537-t003:** The significance of the differences between age groups—results of post-hoc analysis.

Age	Group I	Group II	Group III
**LAS—Benefits** (H (2, *n* = 1025) = 31.45, *p* < 0.01)			
Group I		0.0034 *	0.0001
Group II	0.0034 *		0.0254 *
Group III	0.0001 *	0.0254 *	
**LAS—Involvement** (H (2, *n* = 1025) = 34.74, *p* < 0.01)			
Group I		0.0001 *	0.0133 *
Group II	0.0001 *		0.0075 *
Group III	0.0133 *	0.0075 *	
**LAS—Fears** (H (2, *n* = 1025) = 8.93, *p* = 0.011)			
Group I		0.0095 *	0.1809 *
Group II	0.0095 *		0.9038 *
Group III	0.1809 *	0.9038 *	

* *p* < 0.05.

**Table 4 ijerph-19-01537-t004:** Significance of difference between groups in reference to education level—results of post-hoc analysis.

Education	Primary	Vocational	Secondary	Higher
**LAS—Benefits** (H (3, *n* = 1025) = 23.91, *p* < 0.01)				
Primary		1.0000	1.0000	1.0000
Vocational	1.0000		1.0000	1.0000
Secondary	1.0000	1.0000		0.0002 *
Higher	1.0000	1.0000	0.0002 *	
**LAS—Involvement** (H (3, *n* = 1025) = 10.81, *p* = 0.012)				
Primary		0.8979	1.0000	1.0000
Vocational	0.8979		0.0657	0.3786
Secondary	1.0000	0.0657		0.1709
Higher	1.0000	0.3786	0.1709	
**LAS—Fears** (H (3, *n* = 1025) = 2.48, *p* = 0.48)				
Primary		1.0000	1.0000	1.0000
Vocational	1.0000		1.0000	1.0000
Secondary	1.0000	1.0000		1.0000
Higher	1.0000	1.0000	1.0000	

* *p* < 0.05.

**Table 5 ijerph-19-01537-t005:** Significance of differences between groups in reference to type of place of residence—results of post-hoc analysis.

Type of Settlement	The Country	Small Town	Big Town	City
**LAS—Benefits** (H (3, *n* = 1025) = 6.20, *p* = 0.10)				
The country		0.6976	1.0000	1.0000
Small town	0.6976		1.0000	1.0000
Big town	1.0000	0.2365		0.3403
City	1.0000	1.0000	0.3403	
**LAS—Involvement** (H (3, *n* = 1025) = 55.83, *p* < 0.01)				
The country		1.0000	0.0001 *	0.0001 *
Small town	1.0000 *		0.0001 *	0.0001 *
Big town	0.0001 *	0.0001 *		1.0000
City	0.0001 *	0.0001 *	1.0000	
**LAS—Fears** (H (3, *n* = 1025) = 5.14, *p* = 0.16)				
The country		0.2528	1.0000	1.0000
Small town	0.2528		0.3561	1.0000
Big town	1.0000	0.3561		1.0000
City	1.0000	1.0000	1.0000	

* *p* < 0.05.

**Table 6 ijerph-19-01537-t006:** The significance of differences between groups in reference to type of the house—results of post-hoc analysis.

Type of House	Block of Flats	Multi-Family House	Detached House
**LAS—Benefits** (H (2, *n* = 1025) = 1.99, *p* = 0.37)			
Block of flats		1.000	1.000
Multi-family house	1.000		0.5883
Detached house	1.000	0.5883	
**LAS—Involvement** (H (2, *n* = 1025) = 18.85, *p* < 0.01)			
Block of flats		0.0068 *	0.0003 *
Multi-family house	0.0068 *		0.8314
Detached house	0.0003 *	0.8314	
**LAS—Fears** (H (2, *n* = 1025) = 1.23, *p* = 0.54)			
Block of flats		1.000	0.9391
Multi-family house	1.000		1.000
Detached house	0.9391	1.000	

* *p* < 0.05.

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
