# Peer review of "Sociodemographic Determinants of Poles’ Attitudes towards the Forest during the COVID-19 Pandemic"

_ijerph, 2022, doi:10.3390/ijerph19031537_

Round 1

Reviewer 1 Report

The article deals with an important problem of the social use of forests. Spending time in the woods on a regular basis strengthens the immune system, alleviating the effects of social isolation and loneliness, which have a negative impact on both physical and mental health. What Matters During the Pandemic Period. At the same time, the increase in the number of visitors to forests was a challenge for forest managers and the city's forestry policy.
Reviewer's comments:

1) It seems that the information about the research group in the title should be specified. The journal has an international dimension, therefore it is necessary to select a research group. Information about the research group appears only in the summary - it is not enough. Moreover, it is necessary to consolidate the hypotheses, do they have a thesis dimension or are they research questions?
2) Hypothesis 1 seems to result from the research carried out; similarly hypothesis 5 and 6

Stylistic and punctuation errors:
Line 59 "Korze-niewicz [7]"
Line 70: "benefits"

review of the literature
Add current literature on research conducted in Poland
• Monika Wojcieszak-Zbierska, Anna JÄ™czmyk, Jan Zawadka, JarosÅ‚aw Uglis: Agritourism in the era of coronavirus (COVID-19): Quick assessment from Poland, Agriculture, 2020, 10, 9, pp. 1-19

Jarosław Uglis, Anna Jęczmyk, Jan Zawadka, Monika Małgorzata Wojcieszak-Zbierska, Marcin Pszczoła: The impact of the COVID-19 pandemic on tourism plans: a case study from Poland. Current Problems in Tourism, 2021, pp. 1-16.
Correct references: 15, 23, 28

Author Response

Thank you for review our manuscript. We revised, corrected and suplemented manuscript with suggestions. We hope, that we improved the quality of this manuscript and we have hance to acceptable for publication.

Below we have listed the answers.

  • It seems that the information about the research group in the title should be specified. The journal has an international dimension, therefore it is necessary to select a research group. Information about the research group appears only in the summary - it is not enough. Moreover, it is necessary to consolidate the hypotheses, do they have a thesis dimension or are they research questions?

Thank you for your attention. We corrected the manuscript as suggested and changed the title from The attitude towards the forest during the pandemic. Sociodemographic characteristics to characteristicsSociodemographic determinants of Poles’ atti-tudes towards the forest during the covid-19 pandemic.

  • Hypothesis 1 seems to result from the research carried out; similarly hypothesis 5 and 6

Hypotheses are potential answers to a given research problem. They were formulated on the basis of an analysis of the existing literature research, knowledge of social mechanisms and observation of human behavior in during the pandemic. All the hypotheses were formulated before conducting research. The analysis of the research results allowed for their verification, which was described in the article.

Stylistic and punctuation errors:

Line 59 "Korze-niewicz [7]"

Line 70: "benefits"

Thank you for your attention. We tried to rule out all stylistic and punctuation errors.

review of the literature

Add current literature on research conducted in Poland

Monika Wojcieszak-Zbierska, Anna Jęczmyk, Jan Zawadka, Jarosław Uglis: Agritourism in the era of coronavirus (COVID-19): Quick assessment from Poland, Agriculture, 2020, 10, 9, pp. 1-19

Jarosław Uglis, Anna Jęczmyk, Jan Zawadka, Monika Małgorzata Wojcieszak-Zbierska, Marcin Pszczoła: The impact of the COVID-19 pandemic on tourism plans: a case study from Poland. Current Problems in Tourism, 2021, pp. 1-16.Correct references: 15, 23, 28

We supplemented the manuscript with suggested and current literature.

Reviewer 2 Report

The presented topic of study and justification are timely and very paramount.

The authors did a good job but the manuscript must be improved and can be accepted for publication.

The following could be worked on.

  1. The abstract could be improved i.e include the used methodology, the sample size, and statistical analysis. This matters given such a study.
  2. Use some English editing software to improve some of the sentences as some are hard to read. This will improve the readability. 
  3. The introduction is well used scientific references but these could be supported by most recent studies too. Also, other studies which emphasize the importance of forest recreational services are vital.
  4. The work in the introduction and other parts could be divided into paragraphs for proper and clear readability (24 to 48)!!! This can be divided!
  5. The references cited are insufficient given this particular work. I suggest searching for more relevant studies.
  6. The discussion seems to be narrowed to social demographic characteristics however, more emphasis is placed on the role of forests during the pandemic, and citing more studies that support the notions adds value to this study.

Ceteris paribus, your work is recommendable and will be accepted once the raised issues are addressed.

Author Response

Dear Reviewer,

Thank you for review our manuscript “The attitude towards the forest during the pandemic. Sociodemographic characteristics” to International Journal of Environmental Research and Public Health. We revised and corrected manuscript and we suplemented manuscript with suggestions. 

Below we have listed the answers to the Reviewers.

The following could be worked on.

  1. The abstract could be improved i.e include the used methodology, the sample size, and statistical analysis. This matters given such a study.

Thank you for your attention. We improved and completed the abstract.

  1. Use some English editing software to improve some of the sentences as some are hard to read. This will improve the readability. 

We used some English editing software to improve of the manuscript.

  1. The introduction is well used scientific references but these could be supported by most recent studies too. Also, other studies which emphasize the importance of forest recreational services are vital.

We supplemented the manuscript (in the introduction) with a fairly extensive paragraph on the recreational role of the forest.

  1. The work in the introduction and other parts could be divided into paragraphs for proper and clear readability (24 to 48)!!! This can be divided!

. The chapter is divided into sub-chapters.

  1. The references cited are insufficient given this particular work. I suggest searching for more relevant studies.

The manuscript was supplemented with 29 references.

  1. The discussion seems to be narrowed to social demographic characteristics however, more emphasis is placed on the role of forests during the pandemic, and citing more studies that support the notions adds value to this study.

We tried to complement the discussion, but in this study we focused mainly on people's attitudes to the forest during a pandemic. We have included the most content on the role of the forest in the introduction.

Reviewer 3 Report

Dear Authors,
In your work you have conducted a very extensive research. However, I am not convinced that your analysis and assumptions are correct.
In the introduction, there is a lack of highlighted information concerning lock down restrictions in Poland. Were the forests closed, were there any quantitative or time limitations in using them? Was it possible to get to the forests by public transport, are there forests in cities? 
Much more needs to be said about why the population during a pandemic wants, should rest in forests. There have been many different publications on this topic in recent years that need to appear with you.
In the introduction you have misapplied citations (for position 4 and 8).
In materials and methods, please state what specific statements in each category respondents had to rate. How groups were compared if there were differences in the number of factors in each category.
In the research procedure you write about spreading information about the survey on social media, please state in which media, in which groups.
I strongly disagree with the assumption that a person at 21 is a young person and at 26 is an older person. If this has already been done it should be explained methodological assumptions. Can the groups created in this way be compared with the results of studies by other authors? The interpretation of these results should also be approached with caution.
It is likely that more than 50% of the respondents are students who are economically inactive. This has a significant impact on the interpretation of the results. We cannot conclude that inactive people have more time if they are students.
Please review the bibliography again, there are numerous errors

Author Response

Dear Editor,

Thank you for review our manuscript “The attitude towards the forest during the pandemic. Sociodemographic characteristics” to International Journal of Environmental Research and Public Health. We revised and corrected manuscript and we suplemented manuscript with suggestions. We hope, that we improved the quality of this manuscript and we have hance to acceptable for publication.

Below we have listed the answers to the Reviewers.

In the introduction, there is a lack of highlighted information concerning lock down restrictions in Poland. Were the forests closed, were there any quantitative or time limitations in using them? Was it possible to get to the forests by public transport, are there forests in cities? Much more needs to be said about why the population during a pandemic wants, should rest in forests. There have been many different publications on this topic in recent years that need to appear with you.

Thank you for your attention. We completed the introduction.

In the introduction you have misapplied citations (for position 4 and 8).

Corrected.

In materials and methods, please state what specific statements in each category respondents had to rate.

Materials and methods are supplemented with examples of statements consisting of on individual scales. The entire tool, along with the details of its design, was presented in another publication.

How groups were compared if there were differences in the number of factors in each category.

As indicated in the text: "in order to determine the differences between the particular groups, Mann–Whitney U test and Kruskal-Wallis one-way analysis of variance by ranks were adopted".

In the research procedure you write about spreading information about the survey on social media, please state in which media, in which groups.

Information was transmitted via Facebook, Skype and addresses e-mail (too Messenger, WhatsApp). They were further disseminated by the participants snowball testing.

I strongly disagree with the assumption that a person at 21 is a young person and at 26 is an older person. If this has already been done it should be explained methodological assumptions. Can the groups created in this way be compared with the results of studies by other authors? The interpretation of these results should also be approached with caution.

The division of the respondents into age groups was carried out in accordance with the methods accepted in the social sciences. In order to create groups of similar numbers, we used the percentile method (described in the text), which allows us to select three groups of people: the youngest and the oldest in the study group and those in the middle. Perhaps it was awkward to label these groups "young", "mid-age", and "older", which might suggest age-related terms in general, not just our research sample. It will certainly be more skillful to simply number these groups.

It is likely that more than 50% of the respondents are students who are economically inactive. This has a significant impact on the interpretation of the results. We cannot conclude that inactive people have more time if they are students.

Thank you for this attention. At the hypothesis stage, we assumed that professionally active people will have less time for forest recreation, however, as indicated in the interpretation of the research results, this assumption was not confirmed. Moreover, students in Poland are often professionally active (especially extramural students).

Please review the bibliography again, there are numerous errors.

Thank you for your attention, we corrected that.

Round 2

Reviewer 3 Report

good work congratulations